# Experimental Study to Assess Fracture Toughness in SPS Sintered WC–10% Co Hardmetal by Modifying the Palmqvist Test

**Daniel Willemam Trindade *** 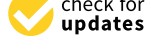, **Renan da Silva Guimarães, Rafael Delorence Lugon, Elias Rocha Gonçalves Junior, Alessandra Agna Araújo dos Santos and Marcello Filgueira**

Advanced Materials Laboratory, Science and Technology Center, State University of Northern Rio de Janeiro Darcy Ribeiro-UENF, 2000 Alberto Lamego Avenue, Campos dos Goytacazes 28013-602, Brazil
* Correspondence: daniel.w.trindade@outlook.com

**Abstract:** Hardmetals are widely used as cutting, machining, and drilling tools for rocks due to their excellent properties of hardness, fracture toughness, and wear resistance over a wide temperature range. This study proposed to evaluate the fracture toughness of WC–10% Co carbide, sintered via spark plasma sintering—SPS, through the Vickers indentation measures, using a modification of the Palmqvist test, which is widely used to assess the toughness of cemented carbides, and to compare this result with the results of six different conventional models: Shetty, Niihara, Laugier, ISO 28079, Hanyaloglu, and Lankford. The model to assess the toughness proposed in this study showed similarity with the Palmqvist test. However, there were considerable differences in the KIC values for the different models, such as 13.36 MPa·m$^{1/2}$ and 4.44 MPa·m$^{1/2}$ for the same application load. Comparing the values of the conventional fracture toughness and proposed fracture toughness, the greatest difference between the fracture toughness values was found in the Lankford equation, which varied by 14.74%. The Hanyaloglu equation showed a smaller difference between the fracture toughness values, with a greater variation of 3.61% and lower variation of 1.54%. Adequate results of hardness were obtained, with a maximum of 20.93 $\pm$ 0.25 GPa, minimum of 15.76 $\pm$ 0.63 GPa, and densification of 99.14 $\pm$ 0.47 g/cm$^3$.

**Keywords:** carbide metal; fracture toughness; Vickers indentation; spark plasma sintering; cemented carbides



## 1. Introduction

Hardmetals (such as WC–Co) are widely used as tools for cutting, machining, and rock drilling because of their excellent properties such as hardness, fracture toughness, and wear resistance over a wide temperature range [1]. They have become the top choice for carbide applications today, with 98% carbide components made from WC–Co [2].

WC–Co is difficult to analyze through conventional mechanical tests, mainly to evaluate its fracture toughness, due to the diversity of equations found in the literature and the emergence of the residual surface stress field arising from sample preparation.

Fracture toughness is an essential parameter for cemented carbide design and performance evaluation applications [2–4]. The main focus of industries is to acquire and maintain fracture toughness and maximize hardness [4].

The fracture toughness of homogeneous WC–Co is extremely important, but there are few methods to assess this property [5]. Palmqvist [6] proposed the quantification of fracture toughness long before the development of mechanical fracture analytical methods, with metal carbide analysis and the use of a Vickers pyramidal diamond indenter to achieve crack patterns. After that, Palmqvist established important variables in the analysis of the fracture process, especially hardness. From this, the determination of fracture toughness ($K_{IC}$) became essential and mandatory in hardmetals research.

Shetty et al. [7] concluded that the cracks generated through the Vickers indenter are of the Palmqvist type in nature, even with the use of a high indentation load, and the typical WC–Co alloys do not present median cracks, while the indenter cracks remain radial of Palmqvist by nature.

Several equations were developed to determine fracture toughness by different authors [8]. Some of these equations are derived from a modified dimensional analysis, based on experimentally developed conversion factors, through occasional vague comparisons to a theoretical basis. In different articles, equations are presented that are manipulations of previously proposed equations but with new calibration constants, in order to obtain reasonably correct values for fracture toughness [9].

Liquid-phase sintering under high sintering temperatures is commonly used for the production of hardmetals [10]. When compared to the conventional sintering process and other methods, the spark plasma sintering (SPS) technique has remarkable advantages such as a high heating rate, a lower sintering temperature, applied pressure, and vacuum condition, and monitored densification behavior [11]. It is worth mentioning that, by using the SPS technique, highly densified structures can be obtained through the simultaneous application of pressure and high heating rate, sintering the structure in a shorter time [12].

The determination of conventional fracture toughness is done in a linear way, measured from the beginning to the end of the crack, although the crack propagates in a nonlinear way. At room temperature, hardmetals show a brittle fracture behavior on the macro level. On the micro level, the brittle fracture of the carbide phase and the ductile fracture of the binder phase are noted [13]. Two-phase cemented carbides have four fracture modes: C-C, intergranular fracture of the carbide phase along the carbide-carbide boundaries; C, transgranular fracture through carbide crystals; B, ductile fracture through the binder phase; B-C, fracture along or close to the carbide-ligand interfaces [14].

This research evaluates the fracture toughness of a WC-based carbide with a 10% Cobalt binder, sintered by SPS, alternatively using a modification of the Palmqvist test, analyzing and measuring the crack considering their entire trajectory, using confocal laser microscopy, and calculating by different equations found in the literature: Shetty et al. [7] (Equation (1)), Niihara [15] (Equation (2)), Laugier [16] (Equation (3)), ISO 28079 [17] (Equation (4)), Hanyaloglu et al. [18] (Equation (5)), and Lankford [19] (Equation (6)). Loads of 2 kgf, 15 kgf, 30 kgf, and 45 kgf were used for hardness measurement. Furthermore, none of the literature explores the results of $K_{IC}$ as proposed in this research.

$$K_{IC} = 0.0889 \, (HW)^{1/2} \tag{1}$$

$$K_{IC} = 0.0246 \, (E/H)^{2/5} \, (HW)^{1/2} \tag{2}$$

$$K_{IC} = 0.015 \, (E/H)^{2/3} \, (a/l)^{1/2} \, (P/C^{3/2}) \tag{3}$$

$$K_{IC} = 0.0028 \, (HW)^{1/2} \tag{4}$$

$$K_{IC} = 1.705 \, (WH)^{0.16} \tag{5}$$

$$K_{IC} = 0.00782 \, (H \, a^{1/2}) \, (E/H)^{2/5} \, (c/a)^{-1.56} \tag{6}$$

where $K_{IC}$ is fracture toughness, and P is the applied load during Vickers hardness testing. W is Palmqvist indentation hardness, E and H are the elastic modulus and Vickers hardness, respectively, c is the dimensions of the indentation cracks, a is the indentation diagonal lengths, l is the crack length, and L is the sum of the crack lengths. The cracks generated in the carbide through Vickers indentation, when analyzed at the micro level, show a great sinuosity in their course, caused by the hard WC grains, ductile phase of the binder, and porosity. However, this branching is not considered by the conventional measurement method.

The novelty is related to the crack-length measurement method. The way it is measured in a conventional method, from the point where it starts to the point where it ends, is different from the measurement method adopted in this work, where each path trav-

eled or changed (occurred deflection) was taken into account in the total measurement of the length of the crack. In the proposed method, all crack paths are considered and measured separately.

As an alternative, Brezinová et al. [20] performed a similar analysis using the Lawn and Fuller (LF) model to determine fracture toughness, which does not include the modulus of elasticity (E)/hardness ratio. The authors did not measure the true E value of the coating, and the use of a representative E value would contribute to the overall error in the Palmqvist calculation.

## 2. Material and Methods

For sample preparation, ultra-fine commercial tungsten carbide supplied by Shanghai Xinglu Chemical Co., Ltd., located in Shanghai, China, with an average particle size of 300–400 nm, and extra-fine Cobalt, supplied by Umicore, with a particle size of 1.2 μm, were used, both commercially pure. The powders were weighed with the nominal composition of WC–10% by mass of Co and were mixed in a high-energy mill, model SPEX 8000 Mixer/Mill. The revolution speed was 1500 rpm for 1 h, with a ratio between the ball mass and the powder mass of 10 to 1 in mass. During the mixing process, the solvent Cyclohexane P.A. ($C_6H_{12}$) was added for the purpose of avoiding the oxidation of the powders and minimizing the temperature rise, since this process generates considerable energy. Finally, the powder mixture WC–10% by mass was obtained after drying in an oven at 60 °C.

For the sintering process, a cylindrical graphite matrix was used, which, after assembly, was placed in SPS sintering equipment, model DR. SINTER SPS 211 LX, at 1200 °C, with a pressure of 40 MPa, with a heating rate of 65 °C/min for 5 min. Cylindrical samples with a height and diameter of 5 and 5.5 mm, respectively, were produced.

The density was measured using Archimedes' principle according to ASTM B962. The metallographic preparation consisted of sanding and polishing using suspensions of diamonds up to a particle size of 1/4 μm.

Compression tests were performed on the Instron-5582, at a speed of 1 mm/min. The test performed was the Brazilian Disk Test, with adopted parameters that were based on the ASTM D3967-16 standard.

The Vickers hardness test standardized by the ASTM E-92-82 standard was performed using a Shimadzu digital microhardness tester, model HMV-2T, manufactured by Shimadzu, located at the Advanced Materials Laboratory, Science and Technology Center, State University of Northern Rio de Janeiro Darcy Ribeiro-UENF, in Campos dos Goytacazes, Brazil, with a load of 2 kgf. The hardness was measurement with loads of 15, 30, and 45 kgf in a Pantec durometer, model RBSM, Panambra. The impressions of the loads and diagonals were analyzed and measured using an Olympus confocal laser microscope, model Lext OLS4000, manufactured by Olympus, located at the Advanced Materials Laboratory, Science and Technology Center, State University of Northern Rio de Janeiro Darcy Ribeiro-UENF, in Campos dos Goytacazes, Brazil.

An analysis of scanning electron microscopy (SEM) equipped with energy dispersion spectroscopy (EDS) made it possible to obtain more detailed information on the microstructural characteristics of the materials and allowed for identifying chemical composition of the materials in specific points and areas, observing the behavior of cracks, and analyzing their path. This analysis was performed using a scanning electron microscope, manufactured by ZEISS, model EVO/MA 10, located at the Metallurgical Technology Sector of the Federal Institute Espírito Santo, in Vitória, Brazil.

*Length of Cracks*

The analyzing conventional method is shown in Figure 1. Where d1 and d2 are the diagonals of the indenter; a is the half of the diagonal impression of the indenter (d/2); c is the size of the surface crack, added to half the diagonal of the indentation, where (c = l + a); and ln is the length of each crack formed.

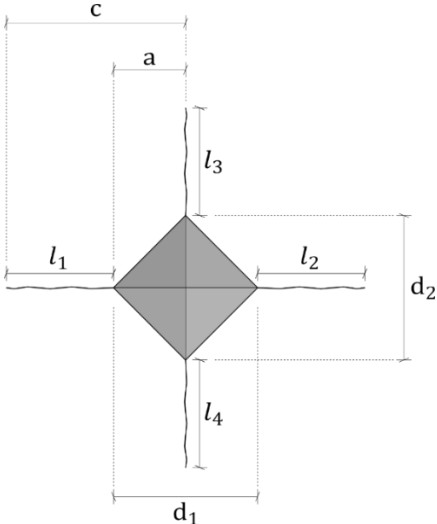

**Figure 1.** Schematic diagram of indentation characteristics.

The cracks generated in cemented carbide through Vickers indentation, when analyzed at the micro level, show a great sinuosity in their path, caused by WC hard grains, ductile phase of the binder, and porosity. However, this path is not considered by the conventional measurement method. Most of the previous fracture toughness studies and their conclusions are based on the conventional method of sintering WC–Co carbide. The pulsed plasma sintering method allows, in short amounts of time, to reach high densifications at comparatively low temperatures, with short sintering times when compared to conventional techniques.

The change in the measurement methodology of the crack lengths generated by the Vickers indenter consisted of considering the crack, measuring all its sinuosity during propagation (Figure 2). Where ln divides, considering all the way the crack has traveled, the difference lies in the way in which the cracks are measured. In the method used, all crack paths are considered and measured separately. Leaving $l_1 = l_1$ (conventional linear) to become $\sum l_1{}^{n'} = l_1{}' + l_1{}'' + l_1{}''' + l_1{}'''' + l_1{}^{n'}$. Likewise, $l_2 = \sum l_2{}^{n'} = l_2{}' + l_2{}'' + l_2{}''' + l_2{}'''' + l_2{}^{n'}$, $l_3 = \sum l_3{}^{n'} = l_3{}' + l_3{}'' + l_3{}''' + l_3{}'''' + l_3{}^{n'}$ e $l_4 = \sum l_4{}^{n'} = l_4{}' + l_4{}'' + l_4{}''' + l_4{}'''' + l_4{}^{n'}$.

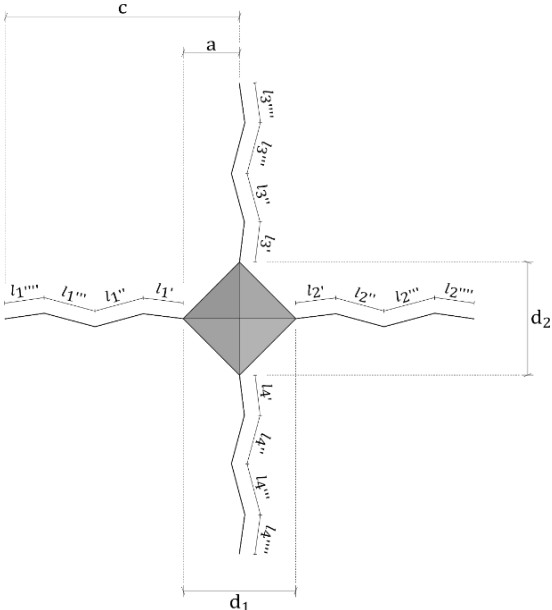

**Figure 2.** Representative model of a Vickers indentation with crack measurement modification.

## 3. Results and Discussions

### 3.1. Characterization of Samples

Figure 3 shows a scanning electron microscopy analysis performed on the sintered sample of the WC–10% Co mixture via SPS at 1200 °C. A good distribution of the Co binder between the WC particles is observed, contributing to a good densification of the structure. This is shown by the dark gray colorings, a greater amount of the Co ligand, and the lighter gray parts of the WC. The samples had a density of 14.51 $\pm$ 0.07 g/cm$^3$ and a densification of 99.14 $\pm$ 0.47%.

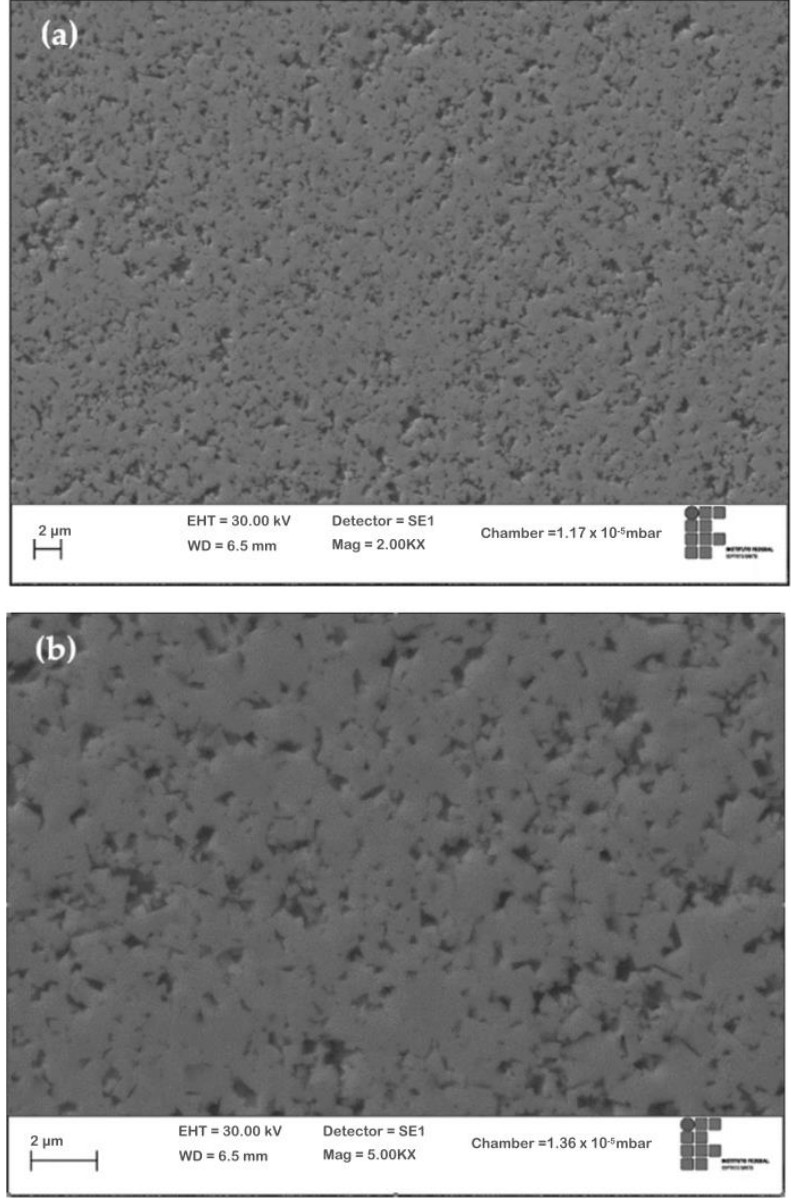

**Figure 3.** WC–10% Co sample sintered via SPS at 1200 °C: (**a**) 2000× magnification and (**b**) 5000× magnification.

A chemical analysis was performed by area mapping on the sintered sample with a magnification of 2000×, performed by SEM along with EDS, as shown in Figure 4. According to the analysis data shown in Figure 4, the sintered sample only presents the elements W (81.7%), C (9.0%), and Co (9.3%).

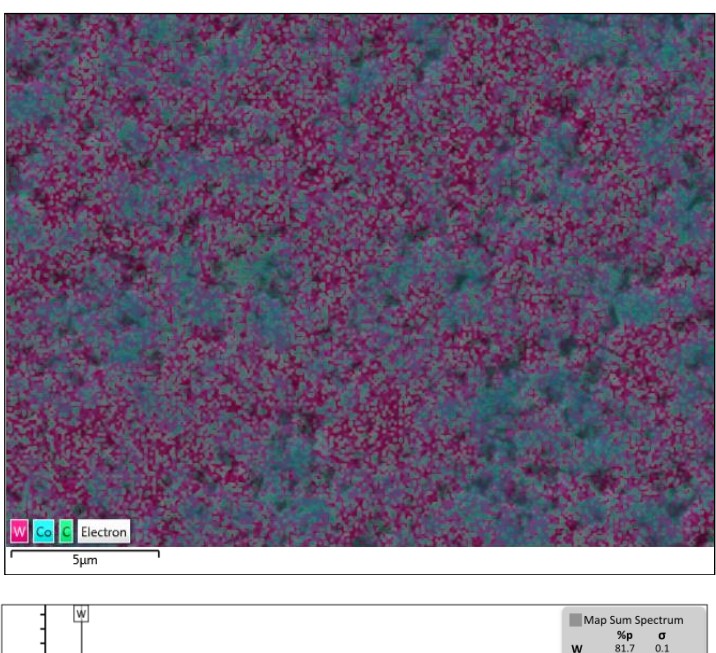

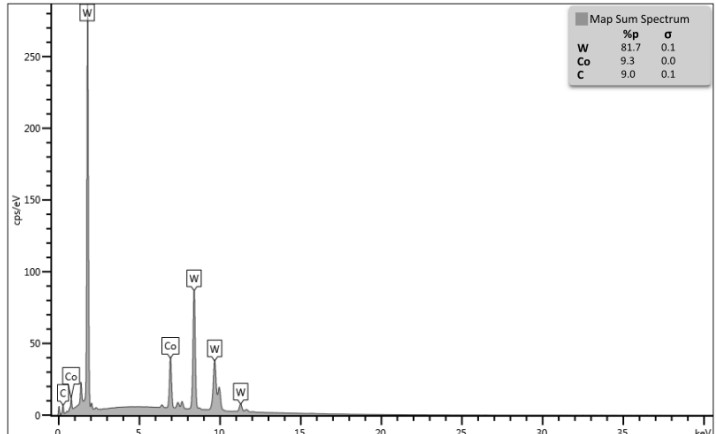

**Figure 4.** Area mapping with EDS spectrum in the WC–10% Co sample sintered via SPS at 1200 °C.

A punctual EDS was performed on the sintered sample to check for the presence of any embrittlement phase. However, no appearance of it was identified in the analyzed region. Figure 5 shows the microstructure of the region analyzed by punctual EDS obtained via SEM with 2000× magnification.

Table 1 shows the results of the punctual semiquantitative analysis performed on the sintered sample, containing the six points analyzed in Figure 5.

**Table 1.** Semi-quantitative analysis, via punctual EDS, of the elements present in the WC–10% Co sample sintered via SPS at 1200 °C.

| Spectrum | Element (%) | | |
|---|---|---|---|
| | W | Co | C |
| 1 | 60.7 | 31.9 | 7.4 |
| 2 | 62.7 | 30.8 | 6.5 |
| 3 | 79.2 | 11.1 | 9.7 |
| 4 | 77.7 | 12.9 | 9.4 |
| 5 | 86.4 | 4.0 | 9.6 |
| 6 | 87.7 | 3.0 | 9.3 |

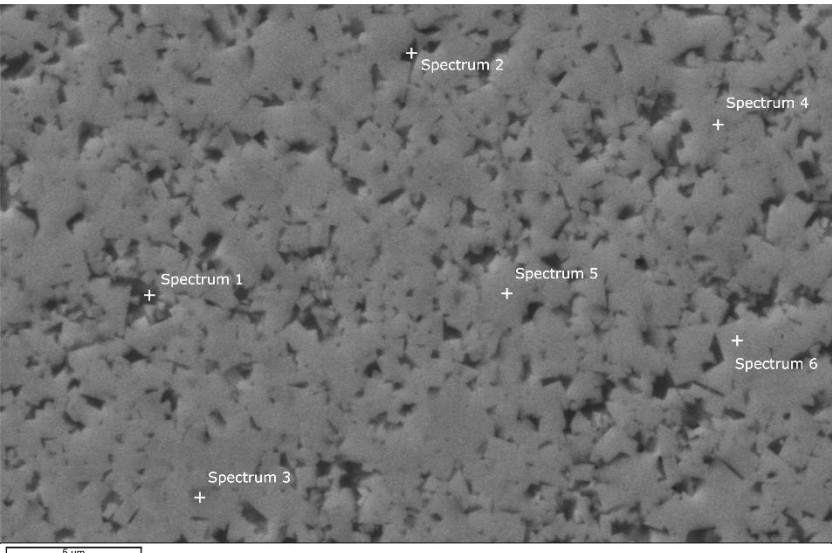

**Figure 5.** Sintered sample region (WC–10% Co) analyzed by punctual EDS.

Analyzing the information presented in Table 1, point 1 and point 2 show a darker gray region with a higher concentration of the Co ligand. Point 3 and point 4 show a region with medium gray color represented by a higher concentration of W and a median concentration of Co. Point 5 and point 6 show a light gray region, where the lowest concentrations of the Co ligand are represented. However, in none of the points analyzed were the amounts of W and Co close or equivalent; it can be concluded that, in principle, there is no indication of the eta phase ($\eta$), which is considered embrittlement to the material.

Figure 6 shows the X-ray spectra of the analyzed region via the punctual EDS of the WC–10% Co composite sintered at 1200 °C via SPS. The points marked on the microstructure are relative to the emission spectra characteristic of X-rays with the same numbering in Figure 5.

Table 2 presents the hardness values calculated with the different applied loads, together with the measurements of the depths of the indentations.

**Table 2.** Hardness of sintered samples WC–10% Co as a function of applied loads.

| Compound | Load Applied (kgf) | HV (kgf/mm$^2$) | Hardness (GPa) | Indentation Depth (µm) |
|---|---|---|---|---|
| | 2 | 2134.30 ± 25.04 | 20.93 ± 0.25 | 4.56 ± 0.17 |
| WC–10% Co | 15 | 1700.74 ± 123.01 | 16.68 ± 1.21 | 17.84 ± 1.25 |
| | 30 | 1663.59 ± 49.65 | 16.31 ± 0.49 | 24.00 ± 0.59 |
| | 45 | 1607.06 ± 64.67 | 15.76 ± 0.63 | 31.03 ± 1.27 |

As described by Fang, Koopman, and Wang [21], the hardness of WC–Co hardmetals varies between 700 and 2200 HV. The results obtained in the sintered samples, for all loads, are according to the expected values for WC–Co hardmetals.

As noted in Table 2, hardness values decrease as applied loads increase. This fact occurs due to the increase in the area reached by the indentation for each load. Since, with each increase, the indentation covers an area with greater defects, a greater set of grains, a greater combination of the WC and Co binder, and a greater porosity. A load of 2 kgf was not enough to generate cracks in the sample under study. Therefore, it was disregarded for the calculation of fracture toughness. Confocal laser microscopy enabled the analysis of the depths of the indentations generated from each load applied to the samples.

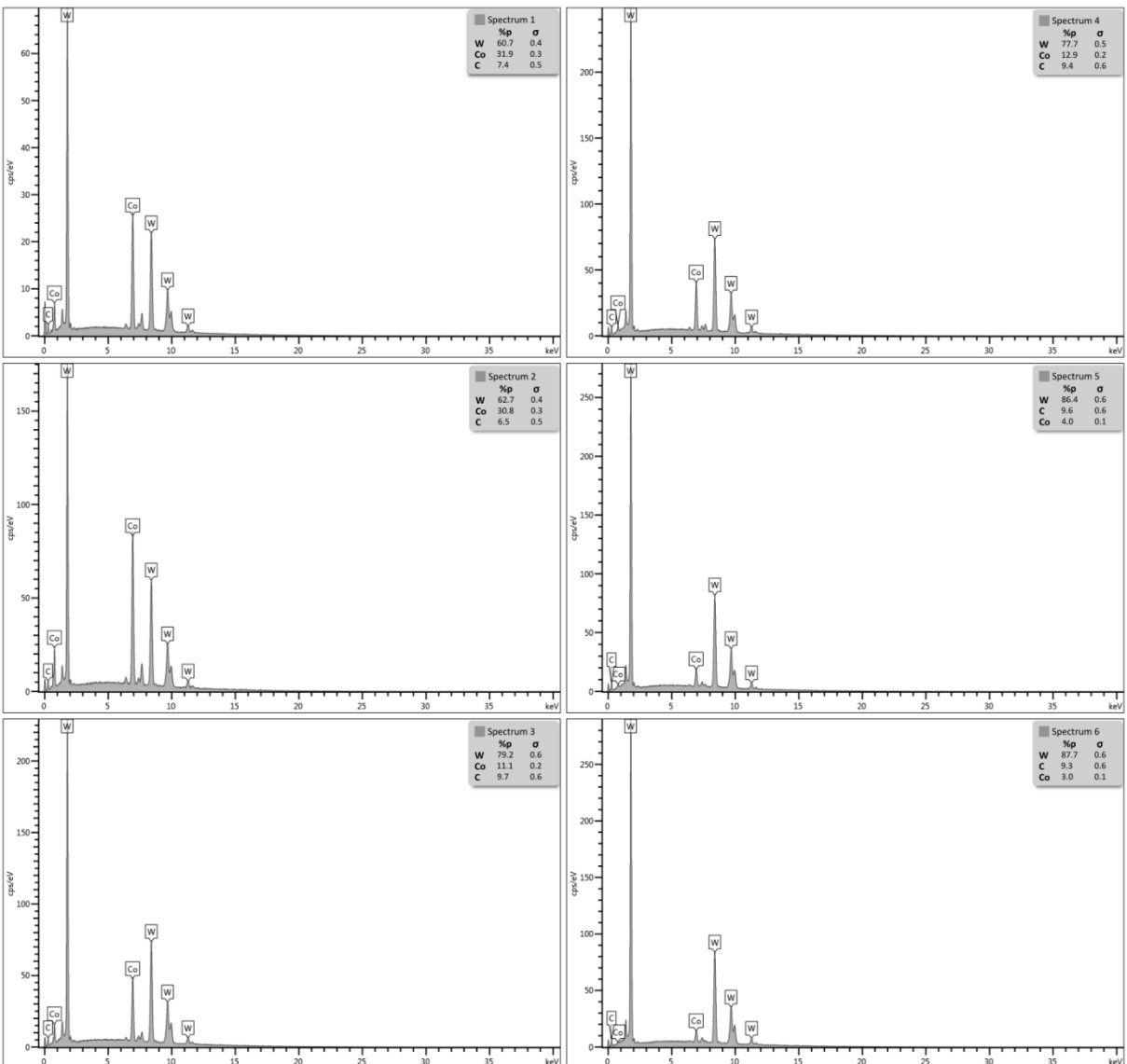

**Figure 6.** X-ray spectra of the analyzed region via punctual EDS of the WC–10% Co composite sintered via SPS at 1200 °C.

### 3.2. Crack-Behavior Analysis

The cracks were analyzed by SEM/EDS, to verify the fracture mode, its behavior, and the trajectory along the propagation.

Figure 7 shows the fracture mode and crack behavior. Applied loads of 15, 30, and 45 kgf were performed by SEM, with a 5000× magnification.

By analyzing the samples, there was a mixed mode of crack propagation, with intergranular (C-C) and transgranular (C) cracks, associated with fractures along, or close to, the carbide–binder interfaces (C-B) and ductile fractures (B) occurring in the binder phase.

For WC–Co carbide, crack deflection, crack bridging, and crack branching have the effect of increasing the fracture areas and consuming more load force energy. It is interesting to note that a crack is impaired when it encounters the thin WC region. The propagation direction changes around the smaller WC grain regions and propagates in the larger WC grain region. As a result, the crack is deflected to a large extent in its trajectory. This phenomenon can be attributed to the fact that the WC region with a larger grain size is more fragile compared to the fine WC aggregate [22].

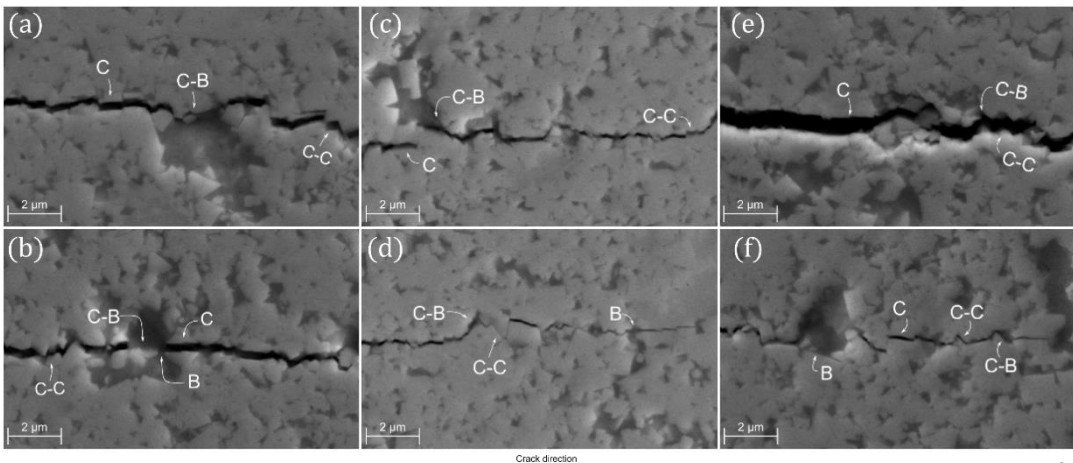

**Figure 7.** Fracture mode in the WC–10% Co sintered sample: (**a**,**b**) load of 15 kgf, (**c**,**d**) load of 30 kgf, and (**e**,**f**) load of 45 kgf.

It is known that fracture toughness derives from the surface area of the crack and from the specific energy of the crack, i.e., the energy required for the crack to develop per unit area. The surface area is given by the length of the crack path within a unit of distance [23].

Figure 8 shows the microscopy made by SEM of the propagation of cracks in the WC–10% Co sintered samples with load application of 15, 30, and 45 kgf, respectively, analyzing the behavior of the cracks.

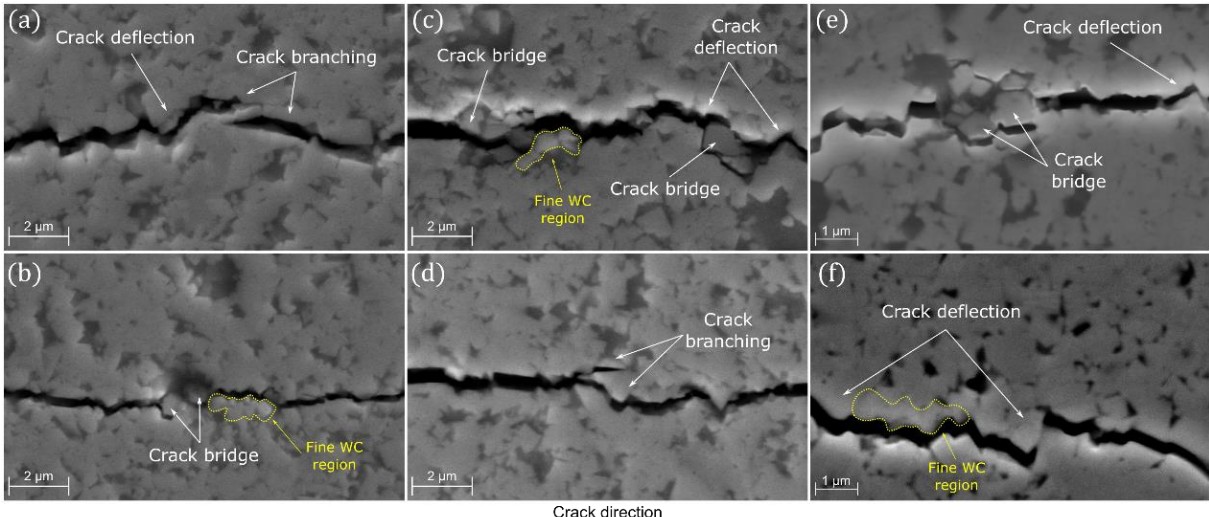

**Figure 8.** Crack behavior in the sintered sample: (**a**,**b**) load of 15 kgf, (**c**,**d**) load of 30 kgf, and (**e**,**f**) load of 45 kgf.

At all loads, the cracks show crack deflection, which is a change in their natural path caused by the hard WC grains, mostly in regions with small grain sizes (yellow circles). This presents crack branching, where the crack splits into two or more parts, propagating in more than one direction. This also shows crack bridges, which are formed by hard WC grains, occurring when the cracks do not have the strength or energy needed to break them. However, they bypass the grains and continue their trajectories.

As shown, the cracks generated in the WC–10% Co composite are not linear, making it pertinent to calculate the fracture toughness in the complete way of the cracks, by considering the entire trajectory.

*3.3. Fracture Toughness Analysis*

Figure 9 shows the crack-measurement method performed in this work. Figure 9a shows a crack generated at one of the four points of the Vickers indentation, with a load of 15 kgf. Figure 9b shows the conventional linear measurement of the total crack length, starting and ending at the yellow points. Figure 9c shows the complete way of measuring the total length of the crack proposed and performed in this work, where the total length is the sum of all the measured lengths, point by point (manually), of each path covered by the crack, considering all linearity variation. The images were obtained by confocal laser microscopy at a magnification of 1000×. The aforementioned magnification was adopted due to equipment limitations.

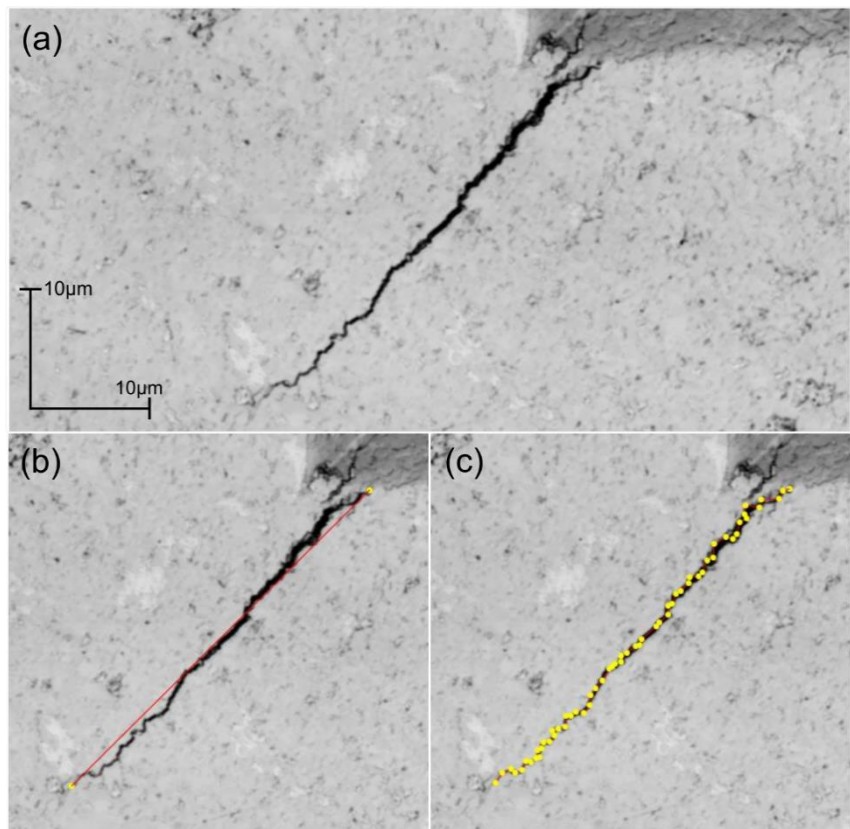

**Figure 9.** Crack measurement type: (**a**) crack generated with 15 kgf, (**b**) conventional linear measurement, and (**c**) proposed modification measurement.

The cracks shown in Figure 10 were obtained with 1000× magnification and 2× digital magnification. The cracks shown in Figure 11 were obtained at 1000× magnification and 1.2× digital magnification, and the cracks in Figure 12 were obtained at 1000× magnification, both by confocal laser microscopy.

There was an increase in the total crack length at all applied loads. The greater the application load was, the greater the crack length. In the proposed measurement, there was an average increase of 42.09 μm for the 15 kgf load, an increase of 55.72 μm for the 30 kgf load, and an increase of 64.91 μm for the 45 kgf load. It can be said that, with each increase in load, the difference in the total length of the cracks increases.

After obtaining the crack length data and the data on the diagonals of indentation and modulus of elasticity, the fracture toughness calculations were performed, using Equations (1)–(6). Table 3 presents the fracture toughness results, showing the results calculated with conventional linear crack measurements and the proposed modification measurement.

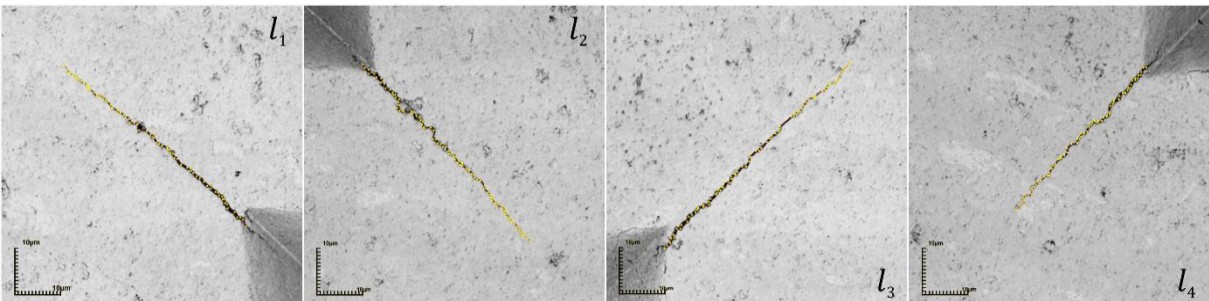

**Figure 10.** Cracks generated with a load of 15 kgf using the proposed measurement.

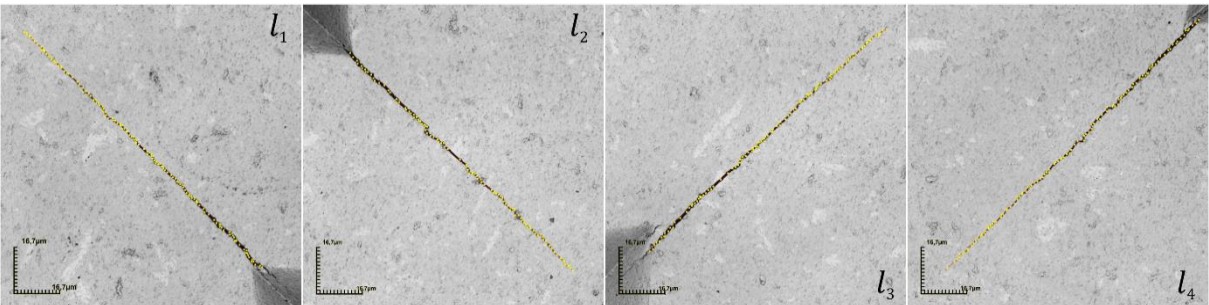

**Figure 11.** Cracks generated with a load of 30 kgf using the proposed measurement.

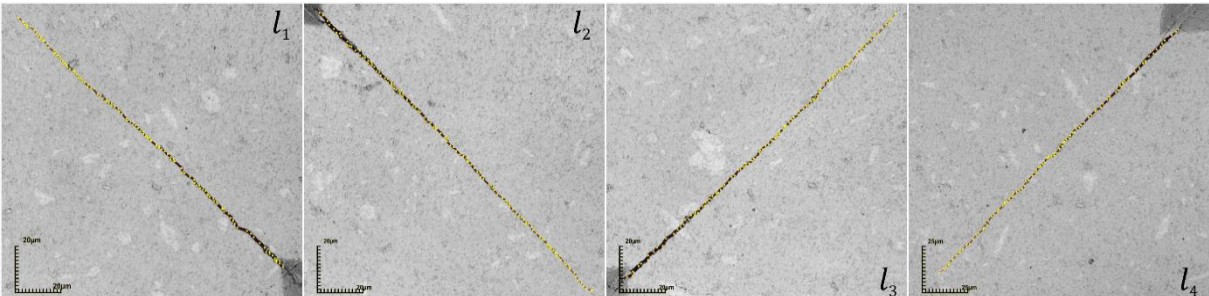

**Figure 12.** Cracks generated with a load of 45 kgf using the proposed measurement.

**Table 3.** Fracture toughness in the WC–10% Co sample.

| | Fracture Toughness (MPa·m$^{1/2}$) | | | | | |
|---|---|---|---|---|---|---|
| **Equation** | **Linear Measurement** | | | **Proposed Measurement** | | |
| | **15 kgf** | **30 kgf** | **45 kgf** | **15 kgf** | **30 kgf** | **45 kgf** |
| Shetty [7] | 10.36 ± 0.14 | 10.87 ± 0.69 | 9.49 ± 021 | 9.23 ± 0.21 | 10.11 ± 0.58 | 9.04 ± 0.22 |
| Niihara et al. [15] | 13.36 ± 0.32 | 14.07 ± 0.98 | 12.31 ± 0.25 | 11.91 ± 0.21 | 13.08 ± 0.81 | 11.72 ± 0.21 |
| Laugier [16] | 4.44 ± 0.32 | 4.01 ± 0.70 | 2.43 ± 0.17 | 3.39 ± 0.16 | 3.34 ± 0.53 | 2.11 ± 0.12 |
| ISO 28079 [17] | 10.32 ± 0.14 | 10.83 ± 0.69 | 9.45 ± 0.21 | 9.20 ± 0.21 | 10.07 ± 0.58 | 9.00 ± 0.22 |
| Hanyaloglu et al. [18] | 11.37 ± 0.05 | 11.54 ± 0.25 | 11.05 ± 0.08 | 10.96 ± 0.08 | 11.28 ± 0.22 | 10.88 ± 0.08 |
| Lankford [19] | 10.72 ± 0.37 | 10.79 ± 1.20 | 8.11 ± 0.33 | 9.14 ± 0.24 | 9.65 ± 1.02 | 7.40 ± 0.25 |

All the values obtained are in agreement with the literature, according to Prakash [24], where the fracture toughness varies from 7 to 25 MPa·m$^{1/2}$, except for the values found with the Laugier equation [16], where the values are below the value expected for WC–Co cemented carbide. It should be noted that there is a considerable variation in the values of fracture toughness, when comparing all the equations used. The lowest values of fracture

toughness were arranged by the Laugier equation [16] and the highest values by the Niihara equation [15].

Large differences in $K_{IC}$ values were observed for different equations. The use of different equations for indentation-fracture toughness calculations showed differences of 33% for the $K_{IC}$ values of the composition studied.

The Hanyaloglu equation [18] was the equation in which there was a greater congruence between the values of fracture toughness calculated in the conventional way and those calculated in the proposed way, and it showed little sensitivity to load variation, remaining at intermediate values compared to other equations; it is, therefore, the most suitable for the carbide WC–10% Co.

There were no large dispersions of results using the current technique for determining fracture toughness and the modification used, especially when using the Hanyaloglu equation, where at loads of 15 kgf the difference between the results was 3.63%. With a load of 30 kgf, it varied by 2.31%. In addition, for a load of 45 kgf, the variation was 1.56%.

Figure 13 demonstrates the variation in fracture toughness with applied loads of 15, 30, and 45 kgf, respectively.

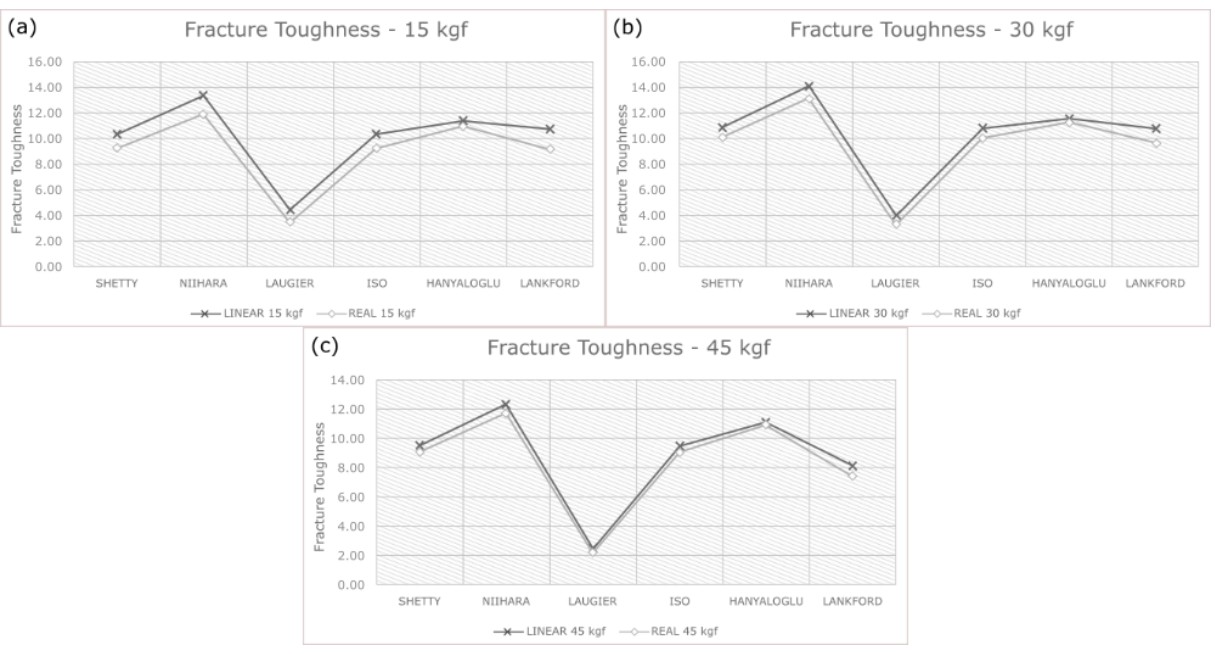

**Figure 13.** Variation of conventional fracture toughness versus proposed fracture toughness: (**a**) load of 15 kgf, (**b**) load of 30 kgf, and (**c**) load of 45 kgf.

Comparing the values of fracture toughness, the greatest variation with a load of 15 kgf (Figure 13a) occurred with the Lankford equation [19], with a drop of 10.72 to 9.14 MPa·m$^{1/2}$, adding up to a difference of 1.58 MPa·m$^{1/2}$. The Shetty equation [7] also obtained a relevant difference, with a decrease from 10.36 to 9.23 MPa·m$^{1/2}$, with a difference of 1.13 MPa·m$^{1/2}$. The smallest difference, 0.41 MPa·m$^{1/2}$, was found by the Hanyaloglu equation [18], ranging from 11.37 to 10.96 MPa·m$^{1/2}$.

With the applied load of 30 kgf (Figure 13b), the greatest difference in fracture toughness was observed for the Lankford equation [19], with a linear fracture toughness value of 10.79, and 9.5 MPa·m$^{1/2}$ for the fracture toughness proposed, indicating a difference of 1.14 MPa·m$^{1/2}$. The Shetty equation [7] showed a difference of 0.76 MPa·m$^{1/2}$ for this load. The smallest difference, 0.26 MPa·m$^{1/2}$, was seen for the Hanyaloglu equation [18].

The variation of fracture toughness for the load of 45 kgf (Figure 13c) was lower than for the load of 30 and 15 kgf. The biggest difference being 0.71 MPa·m$^{1/2}$, noted for the Lankford equation [19], followed by the Niihara equation [14] with a decrease of

0.59 MPa·m$^{1/2}$. The Shetty equation [7] obtained a reduction of 0.45 MPa·m$^{1/2}$. The smallest difference was for the Hanyaloglu equation [18], with a reduction of only 0.17 MPa·m$^{1/2}$.

Observing the results of fracture toughness in all equations, the equation provided by the ISO 28079 [17] standard remained very close to the equation developed by Shetty [7], maintaining very similar values for all applied loads. The two equations showed congruent results with those found in the literature.

The equation developed by Niihara and collaborators [15] underestimates the value of fracture toughness, as it depends on the material's modulus of elasticity, a variable measure that significantly interfered with their results.

The results obtained with the Laugier equation [16] were well below the expected value for hardmetals, with a significant difference in relation to the other equations demonstrated, and were unable to determine the fracture toughness in WC–Co hardmetals.

The Hanyaloglu equation [18], despite not having been developed specifically for WC–Co carbide, maintained a good relationship between the resulting values at all application loads. Compared with the method proposed in this work, it was the equation that showed the least difference in the values of fracture toughness, between conventional fracture toughness and the proposed fracture toughness, with the greatest difference being 0.41 MPa·m$^{1/2}$, varying from 11.37 to 10.96 MPa·m$^{1/2}$, which was found with the application of a load of 15 kgf. With a load of 30 kgf, the difference was 0.27 MPa·m$^{1/2}$, ranging from 11.54 to 11.28 MPa·m$^{1/2}$, and, with a load of 45 kgf, the difference was 0.17 MPa·m$^{1/2}$, ranging from 11.05 to 10.88 MPa·m$^{1/2}$.

The results found through the Lankford equation [19] remained within the expected values for hardmetals, and their results were the ones that showed the most variations from the conventional measurement and actual measurement of fracture toughness, with a decrease of 10.72 to 9.14 MPa·m$^{1/2}$, varying by 1.58 MPa·m$^{1/2}$. However, it is also an equation dependent on the elastic modulus of the material.

Based on the aforementioned differences, it can be stated that by increasing application load, the difference between the value of conventional fracture toughness and the proposed fracture toughness becomes smaller. However, the difference in the fracture toughness value still exists, regardless of the application load.

## 4. Conclusions

The use of spark plasma sintering allowed for good densification in the sintered samples, contributing to all the analyzes performed on the examined composite. At first, by mapping analysis via SEM/EDS, no eta phase (η) was found in the samples sintered via SPS at 1200 °C.

The Vickers indentation technique is suitable for hardness measurements on WC–10% Co, where the minimum values of 15.76 ± 0.63 GPa and maximum values of 20.93 ± 0.25 GPa were within the range of the values obtained in the literature.

Large differences in K$_{IC}$ values were observed for different equations. The use of different equations for indentation-fracture toughness calculations showed differences of 33% for the K$_{IC}$ values for the studied composition. According to this work, in the Hanyaloglu equation [18], there was a greater congruence between the values of the fracture toughness calculated in a conventional way and those calculated in the proposed way, which presented little sensitivity to load variation and remained in intermediate values compared to other equations. Therefore, it is the most suitable for WC–10% Co cemented carbide. It is worth noting that the modified method can and should be used to measure the length of cracks in other materials where more ramifications and crack deflections are observed, to verify its effectiveness.

This study is relevant, as it proposes to analyze the fracture toughness of conventional WC–Co carbide, with a modification of the analysis methodology for measuring the total length of cracks, generated by the Vickers indenter, using a pulsed plasma sintering technique. Moreover, none of the literature explores the results of K$_{IC}$ as proposed in this work.

In this way, this study indicated that the influence of the crack-length measurement mode did not present significant results for the conventional method to be changed. Therefore, the conventional crack measurement method for determining fracture toughness in hardmetals can, and should, continue to be used.

**Author Contributions:** Conceptualization, D.W.T. and M.F.; methodology, D.W.T. and M.F.; software, D.W.T.; validation, D.W.T.; formal analysis, D.W.T., R.d.S.G. and R.D.L.; investigation, D.W.T. and M.F.; resources, D.W.T., R.d.S.G. and R.D.L.; data curation, D.W.T.; writing—original draft preparation, D.W.T.; writing—review and editing, D.W.T., E.R.G.J., A.A.A.d.S. and M.F.; visualization, D.W.T.; supervision, D.W.T. and M.F.; project administration, D.W.T. and M.F.; funding acquisition, M.F. All authors have read and agreed to the published version of the manuscript.

**Funding:** This research was funded by Capes and CNPq—88882.449547/2019-01.

**Institutional Review Board Statement:** Not applicable.

**Informed Consent Statement:** Not applicable.

**Data Availability Statement:** Not applicable.

**Acknowledgments:** The authors thank Capes and CNPq for the financial support.

**Conflicts of Interest:** The authors declare no conflict of interest. The authors have no financial or proprietary interests in any material discussed in this article. This article is derived from the master's thesis of D.W.T., which was presented in 2021 at the Science and Technology Center, part of the Advanced Materials Laboratory, State University of Northern Rio de Janeiro Darcy Ribeiro—UENF. There are no conflicts of interest between the pre-print of this thesis and this article.

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
