# Peer review of "Experimental Study to Assess Fracture Toughness in SPS Sintered WC–10% Co Hardmetal by Modifying the Palmqvist Test"

_coatings, doi:10.3390/coatings12121809_

Round 1

Reviewer 1 Report

Review of Coatings-2041226 – Experimental study of a new method to assess fracture toughness in SPS sintered WC-10% Co hard metal by modifying the Palmqvist test

General Comments

This paper presents results about using modified Palmqvist test to obtain fracture toughness values (KIC) for WC-10%Co carbide that were developed by Spark Plasma Sintering. The obtained values were compared with several other models, and it was found out that no major difference in the fracture toughness values was observed after considering the modified method of measuring the crack length.

The reviewer believes that the concept has been explained clearly. Furthermore, high-quality SEM images, even at high magnification scales (5,000 X), were taken to use for crack length measurement, which is a great advantage of this work. However, the importance of the study, the reason for selection of this specific material, and above all, the novelty were not explained clearly. That said, the reviewer believes that this paper requires major revisions.

Introduction

i.                    The importance of the studied material and the reason for selecting it for this study must be explained clearly here.

ii.                  The quality if Equations (1-7) is not acceptable. Please do not use images for equations and rewrite them for higher quality.

iii.                Please rewrite line 75 as it is not written properly in its current form.

iv.                The novelty of this research (lines 76-77) is not clearly explained. Is the novelty for the comparison of the results from these equations, or is it related to the method of measuring the crack length, or is it related to this specific material?

Materials and Methods

i.                    The quality of the equations shown in lines 134 – 136 is not acceptable. Please rewrite the equations.

ii.                  It must be explained in this section how the crack length is measured in the modified method.

Results and Discussions

i.                    Lines 199 – 200 and Figure 5(c) has provided some information about the modified method of measuring the crack lengths. It is not clear whether the yellow dots in Figure 5(c) were selected manually or by using an image processing software.

ii.                  In Figure 5(c), some dots are selected in zig-zag fashion along a straight crack, which does not seem necessary. Please explain clearly how these dots were selected and what is the criteria for measuring its accuracy and consistency from one crack to another crack.

iii.                The explanation provided in lines 241 – 257 and lines 273 – 277 are repetitive from Table 2 and Figure 9 and does not add to the knowledge of the readers. Instead of the difference in absolute values, it might be better to just mention the differences in percentage so that the change made by the modified method can be comprehended.

iv.                Please rewrite line 262 as it is not written properly in its current form.

v.                  It may not be a good idea to only discuss the change by using the modified crack length measurement method on just ONE specific material. Comparison of the results obtained for another material or a different percentage of Co, say WC-20%Co carbide, can bring about more interesting results.

Conclusions

i.                    Lines 312 – 313 indicates that not much difference in results was obtained from this modification. This is probably because of the selected material in this study. The modified method for measuring the crack length may be more important for other materials where more crack branching and deflections are observed. Again, the reviewer believes that considering just ONE material for this study cannot lead to a strong conclusion and selection of at least one other material or composition is strongly recommended.

Author Response

Introduction

All modifications were carried out as recommended.

Materials and Methods

All modifications were carried out as recommended.

Results and Discussions

All modifications were carried out as recommended.

Conclusions

All modifications were carried out as recommended.

Reviewer 2 Report

Comments to the Authors:
The authors of this paper
evaluate the fracture toughness of WC-based carbide with 10% Cobalt binder, sintered by SPS, alternatively using a modification of the Palmqvist test. It is an interesting work, nevertheless, some details should be considered by the authors:

COMMENT: Page 1, line 31: more references could be added.

COMMENT: Page 8, lines 229-234: The discussion is rather limited. The results presented in Table 2 could be farther commented.

GENERAL COMMENT: Although a large amount of data and results are presented in this article, the discussion is rather limited. More comments could be added.  

The results support the authors’ conclusion. Thus, I think that this paper may be published.

Author Response

All modifications were carried out as recommended.

Reviewer 3 Report

The results of the article are very useful for science and especially for practice, but there are some questions:

1. Most grades of WC-Co alloys are characterized by a certain degree of bimodality of microstructures. At the same time, there is both a fine-grained and coarse-grained fraction in the microstructure. In many cases, this microstructure makes it possible to achieve an optimal combination of hardness and fracture toughness. There is no information on the microstructure of SPS sintered WC-10% Co hardmetal in this article. This does not allow to fully evaluate the physical and mechanical correctness of the obtained results.

2. The title of the article is "Experimental study of a new method to assess fracture toughness in SPS sintered WC-10% Co hardmetal...", but I do not see signs of a new method in the article. What do the authors see as the novelty of the assess fracture toughness method?

3. The authors of the article proposed a new crack measurement method. But it will probably be of interest to the authors to know that similar methods have already been proposed before:

https://link.springer.com/article/10.1007/s11003-009-9144-0

https://link.springer.com/article/10.3103/S8756699011040066

How is the method described in this article better than the known ones?

What is the error of the method?

4. 5. The author proposed a new method that:

- more precisely than the known ones, then by how %?

- physically more correct?

- less labour intensive?

5. To determine KIc, the authors used a number of formulas. These are empirical dependencies based on both positive aspects and limitations. At the same time, these dependencies initially (when they were written) had different accuracy. To what extent is it correct to compare them in this case? After all, their different sensitivity to the structure of this material is already intuitively understandable.

6. What is the main value for science of this article? I propose to write this in the conclusions.

7. Methods are known for the direct determination of fracture toughness in the case of three-point bending. Why the authors did not verify the obtained results with the data of the direct fracture experiment.

Author Response

  1. The microstructure of SPS sintered WC-10% Co carbide was provided along with mapping analysis and EDS point analysis.
  2. Inserted in the last paragraph of the Introduction section: "The novelty is related to the crack length measurement method. The way it is measured in a conventional method, from the point where it starts to the point where it ends, and the measurement method adopted in this work, where each path traveled or changed (occurred deflection) was taken into account in the total measurement of the length of the crack. In the proposed method, all crack paths are considered and measured separately.".

3. 4. A new method was not developed, but a comparative study between the conventional way of measuring the total length of the crack and the measurement considering the alternate paths taken by them during their formation. Therefore, we made a modification to the title of the paper.

5. Their comparison is correct, as they are all used to determine fracture toughness in hard metals. However, they differ in results, as each one has its variation of coefficients, precisely because they were developed empirically by several researchers.

6. Inserted in the last two paragraphs of the Conclusion section: "This study is relevant, as it proposes to analyze the fracture toughness of conventional WC-Co carbide, with a modification of the analysis methodology for measuring the total length of cracks, generated by the Vickers indenter, and using a pulsed plasma sintering technique. Besides, there is no literature exploring the results of KIC as proposed in this work. In this way, this study indicated that the influence of the crack length measurement mode did not present significant results for the conventional method to be changed. Therefore, the conventional crack measurement method for determining fracture toughness in hardmetals can, and should, continue to be used.".

7. The evaluation method of the fracture toughness of ceramic materials involves using a Vickers diamond indenter to induce radial cracks in the material. This method has been considered an attractive method due to the ease and low cost of conducting experiments.

All modifications were carried out as recommended.

Round 2

Reviewer 1 Report

It appears that there is a scaling issue with Figures 3 and 4. Please resolve the issue before the final submission. The content has improved nicely, and the reviewer believes that the manuscript would be ready for publication after making the required modification.

Author Response

(The authors gave the same response as above.)

Reviewer 3 Report

The article has been amended, but some shortcomings still need to be corrected:

1. As a separator of fractional numbers, I recommend using not a comma ",", but a dot ".", (in tables 2,3).

2. In the article https://www.mdpi.com/2075-4701/10/12/1675, there is another formula for determining fracture toughness, KIc, for namely WC-Co alloys. I propose to mention it in the introduction and explore it in the article.

3. The authors, unfortunately, ignored my question about accuracy, but it is important. I propose to look deeper into this issue. After all, if we consider the measurement of hardness only as a macroparameter, then taking into account the branching of the crack will be incorrect, because a change in the curvature of the front at a point is a local microlevel process. Thus, the authors need to decide on the systematization of measurement at macro- and microlevels. I invite the authors to read the article https://link.springer.com/article/10.1007/s11015-013-9680-6. This will help them justify whether this method requires detailing the local geometry of the crack front or will only lead to additional complications.

4. How was the magnification chosen at which images of Figs 9-12 were obtained? After all, if you choose a larger magnification, the crack front will be more tortuous, and if you choose less, it will be less. This methodological aspect will affect the accuracy of measurements.

Author Response

  1. All modifications were carried out as recommended.
  2. All modifications were carried out as recommended.
  3. All modifications were carried out as recommended.
  4. All modifications were carried out as recommended.

Round 3

Reviewer 3 Report

Accept.